# Numerical Analysis on Natural Convection Heat Transfer in a Single Circular Fin-Tube Heat Exchanger (Part 1): Numerical Method

**DOI:** 10.3390/e22030363

**Published:** 2020-03-21

**Authors:** Jong Hwi Lee, Jong-Hyeon Shin, Se-Myong Chang, Taegee Min

**Affiliations:** 1Department of Mechanical Engineering, Kunsan National University, Gunsan, Jeonbuk 54150, Korea; jhl@kunsan.ac.kr; 2G&D Co., Gunsan, Jeonbuk 54001, Korea; alqpzm802@naver.com; 3R&D Center, S&H Co. Ltd., Suwon, Gyeonggi 16643, Korea; tmin@e-snh.co.kr

**Keywords:** natural convection, circular fin-tube, heat exchanger, numerical method

## Abstract

In this research, unsteady three-dimensional incompressible Navier–Stokes equations are solved to simulate experiments with the Boussinesq approximation and validate the proposed numerical model for the design of a circular fin-tube heat exchanger. Unsteady time marching is proposed for a time sweeping analysis of various Rayleigh numbers. The accuracy of the natural convection data of a single horizontal circular tube with the proposed numerical method can be guaranteed when the Rayleigh number based on the tube diameter exceeds 400, which is regarded as the limitation of numerical errors due to instability. Moreover, the effective limit for a circular fin-tube heat exchanger is reached when the Rayleigh number based on the fin gap size (${\mathrm{Ra}}_{s}$) is equal to or exceeds 100. This is because at low Rayleigh numbers, the air gap between the fins is isolated and rarely affected by natural convection of the outer air, where the fluid provides heat resistance. Thus, the fin acts favorably when ${\mathrm{Ra}}_{s}$ exceeds 100.

## 1. Introduction

Heat exchangers are widely used in homes and industries, as well as in the transportation sector. They are used in various applications, such as air-conditioning, power plants, mechanical facilities, automobiles, and the marine industry. Generally, heat exchangers increase the surface area to improve the heat transfer performance via fins attached on a pipe. One of the representative models is the circular fin-tube heat exchanger, which is primarily used for forced convection as well as natural convection depending on requirements such as durability, and limitations such as dangerous elements. Beyond its importance in the engineering field, it is important to analyze natural convection around a geometrical body to understand the physics of convective heat transfer.

Various studies have been conducted on bodies submerged in fluid with regard to the Nusselt number (Nu), which represents a function of the Rayleigh number (Ra) and the Prandtl number (Pr). For instance, Merk and Prins [1] proposed a normalized correlation for a cylinder or sphere as follows:(1)$${\mathrm{Nu}}_{D}=C{\mathrm{Ra}}_{D}{}^{\frac{1}{4}},\;\;C=0.436,\;\mathrm{Pr}=C_{p}\mu /k=0.7.$$

Morgan [2] suggested that heat transfer for a circular tube can be expressed as follows: (2)$${\mathrm{Nu}}_{D}=C{\mathrm{Ra}}_{D}{}^{n},$$
where C=0.85 and n = 0.188 in the range of $10^{2}{\;<\;\mathrm{Ra}}_{D}<10^{4}$.

Churchill and Chu [3] also proposed a convective heat transfer correlation at the regime of laminar and turbulent flow for a horizontal circular tube (Equation (3)).
(3)$${\mathrm{Nu}}_{D}={\left\{0.6+\frac{0.387{\mathrm{Ra}}_{D}^{\frac{1}{6}}}{{\left[1+{\left(\frac{0.559}{\mathrm{Pr}}\right)}^{\frac{9}{16}}\right]}^{\frac{8}{27}}}\right\}}^{2}.$$

Qureshi and Ahmad [4] analyzed the characteristics of flow at a Pr value of 0.7. They established a correlation for a wide range of mean Nu (from $10^{-2}$ to $10^{7}$), thus solving the Navier–Stokes and energy equations for natural convection around a horizontal cylinder. Abu-Hijleh [5] researched the optimization of the numbers, sizes, and positions of fins to maximize the heat transfer of natural convection in the laminar flow range with one or more high-conductivity fins attached along the circumference in the horizontal direction. Hisham et al. [6] studied the effect of surface roughness on the natural convection of a horizontal cylinder by experimenting on four kinds of models with sandpaper covering the cylindrical surface. They considered Nu to be a function of surface roughness as well as Ra, and showed that the heat transfer can be improved by a maximum of 30% if the contact and sandpaper resistances are considered. Chen and Hsu [7] experimented on various fin pitches for a circular fin-tube heat exchanger and proposed a numerical inverse scheme to predict the mean heat transfer coefficient and fin efficiency using temperature data at six measurement points and the least-squares method. Kang and Jang [8] also conducted several experiments to obtain a correlation between Nu and Ra for various fin diameters and pitches as a function of the diameter ratios and pitch-to-tube diameter. Chen et al. [9] used both numerical and experimental methods to generate correlations for various tube diameters and fin pitches in a vertical flat-plate fin-tube heat exchanger. Recently, Kang and Chang [10] experimentally investigated the natural convection in a circular fin-tube heat exchanger, and proposed an empirical correlation for 16 models of fin-tube diameter ratios (*D_o_/D*) of 1.2 to 2.8, and fin pitch to tube diameter ratios (*s/D*) of 1.2 to 2.6.

Although such experiments helped improve our understanding of the physics of heat transfer, measurement errors are common due to issues such as instrument accuracy and response speed. Unlike general thermal-fluid experiments, far more caution must be exercised as environmental uncertainties may greatly affect the data, and also because we cannot perfectly exclude the heat loss at the temperature sensors and the test section. The effect of radiation emissivity is estimated within 4% in the experiment and should be corrected [10]. Numerical methods have substituted or compensated for the shortcomings of these kinds of natural convection experiments, such as errors, number of man hours, and engineering price. In this research, unsteady three-dimensional incompressible Navier–Stokes equations are used directly in numerical simulations, as in Reference [10], with the Boussinesq approximation used to validate the proposed experimental results for a fin-tube heat exchanger design. The Boussinesq approximation is a method of solving non-isothermal flow such as the natural convection problem, without the need to solve for the fully compressed formula of the Navier–Stokes equations. It assumes that the change in density has no effect other than generating buoyancy in the flow field.

## 2. Numerical Simulation

### 2.1. Equations Governing the Circular Fin-Tube Heat Exchanger

The schematic of the circular fin-tube heat exchanger used in this study is shown in Figure 1. The notations are explained below.

*D*: diameter of the circular tube,

*D_o_*: diameter of the circular fin,

$P_{f}$: fin pitch,

t: fin thickness, and

s: fin gap size, where $s=P_{f}-t$.

The dimensions are given in Table 1.

Considering the heat exchanger as a lumped heat capacity system, its temperature (T) is expressed in the energy equation as follows:(4)$$\frac{dT}{d\tau }+\frac{hA}{\rho c_{p}V}\left(T-T_{f}\right)=0,$$

The convective heat transfer coefficient h can be calculated for the lumped cooling temperature as follows [10]:(5)$$h=-\frac{\rho c_{p}V}{A\Delta \tau }\ln\frac{T-T_{f}}{T_{i}-T_{f}},$$
where ρ is the density of the material, $c_{p}$ is the specific heat, Δτ is the time difference, V is the volume of the system, A is the total surface area bounding to the outer environment, and subscripts i and f denote the initial condition and final state, respectively.

Therefore, the volume (V), area (A), and reference length (L) are defined using simple geometry as
(6)$$V=\frac{\pi }{4}(D_{o}^{2}-D_{hole}^{2})P_{f}+\frac{\pi }{4}(D_{o}^{2}-D^{2})t,\;$$
(7)$$A=\frac{\pi }{2}(D_{o}^{2}-D^{2})+\pi Ds+\pi D_{o}t,\;\mathrm{and}$$
(8)$$L=\frac{\pi \left(D_{o}+D\right)}{4},$$
where $D_{hole}$ is the inner diameter of the tube.

The film temperature ($T_{m}$) and expansion coefficient (β) are defined to obtain the physical properties of air ($T_{\infty }≅T_{f}$) as follows:(9)$$\Delta T=T-T_{f},$$
(10)$$T_{m}=\frac{T+T_{f}}{2}=T_{f}+\frac{\Delta T}{2},\;\mathrm{and}$$
(11)$$\beta =\frac{1}{T_{m}}.$$
The dimensionless parameters, Ra and Nu, are defined as
(12)$${\mathrm{Ra}}_{L}=\frac{g\beta \Delta TL^{3}}{\alpha \nu }$$
(13)$${\mathrm{Nu}}_{L}=\frac{hL}{k}$$
where g is the gravitational acceleration, $\alpha =k/\left(\rho c_{p}\right)$ is the thermal diffusivity, ν is the kinematic viscosity of air, and k is the heat conductivity of the fin material. *L* in Equations (12) and (13) is defined in Equation (8) and can be substituted for other parameters such as fin gap (s) and tube diameter (*D*).
(14)$${\mathrm{Ra}}_{s}=\frac{g\beta \Delta Ts^{3}}{\alpha \nu }.$$
(15)$${\mathrm{Ra}}_{D}=\frac{g\beta \Delta TD^{3}}{\alpha \nu }.$$

Thus, the following equation can be derived from Equations (14) and (15):(16)$${\mathrm{Ra}}_{s}={\mathrm{Ra}}_{D}{\left(\frac{s}{D}\right)}^{3}.$$

### 2.2. Numerical Simulation Method

For the numerical simulation in this research, a three-dimensional laminar incompressible flow was assumed to study the physics of natural convection. The commercial code ANSYS CFX 18 [11] was used in the computation, assuming an unsteady setup for a total time of 3000 s and a time step of 0.01 s. The time step was determined by considering the Courant number (Co≤1).

The computational domain for a single circular fin-tube exchanger is shown in Figure 2a. The domain spans 20 and 30 times the tube diameter (D) in the *x* and *y* directions, respectively, and one pitch ($P_{f}$) in the *z* direction. For both boundaries, at $z=0,P_{f}$, a periodic boundary condition was applied for the fully developed flow in that direction, and the remaining four surfaces were applied with opening condition. Therefore, we must consider a sufficiently wide domain of 20D×30D in the xy plane. 

The air temperature ranged from 19 to 25 °C (room temperature), and the wall temperature was set to a fixed initial value of 50–60 °C. The physical properties of air were given as a function of temperature at standard atmospheric pressure, namely at 0, the density was 1.29 kg/m^3^, the constant pressure heat capacity was 1004.4 J/(kg·K), thermal conductivity was 0.025 J/(m·K), and dynamic viscosity was $1.71\times 10^{-5}$ kg/(m·s). Pr was 0.687.

The grids seen in Figure 2b,c were constructed with tetrahedral unstructured meshes. The size of grid used was approximately 0.002 m. Prism meshes were used near the wall boundaries of the circular tube and fins, and the total number of elements amounted to approximately 0.5 million with 15,000 nodes. The assessment of convergence was conducted by assuming that the relative residual of all the computational variables is less than $10^{-6}$, and that the balance of momentum and energy is 99.9%. A PC with a CPU of 3.0 GHz and 32 GB RAM was used for the computations. The computation time was approximately 72 hours per case.

## 3. Validation of the Computational Fluid Dynamics Method

### 3.1. Circular Tube

To validate the proposed numerical method, a simulation with a simple circular tube was performed to compare with other empirical correlations, namely Equations (1)–(3). Ra based on the tube diameter (${\mathrm{Ra}}_{D}$) ranges from 150 to 10,000, and the flow characteristics are still considered to be laminar for this case. One unsteady sweeping was conducted for Equation (5) to obtain Nu based on the tube diameter (${\mathrm{Nu}}_{D}=\mathrm{hD}/k$), and its corresponding ${\mathrm{Ra}}_{D}$ was computed using ΔT in Equation (15). In Figure 3, the numerical result is compared with the correlations seen in Equations (1)–(3) as well as an unpublished experimental result produced by Kang. Kang’s experimental data were obtained with the same experimental method proposed in Reference [10]. The heat transfer of the circular tube can be categorized using the ranges ${\mathrm{Ra}}_{D}=400$, $150<{\mathrm{Ra}}_{D}<400$, and $400<{\mathrm{Ra}}_{D}<10^{4}$. The low Ra region shows oscillations because the temperature difference is too small to measure the error that may affect Nu. Thus, a better-controlled experiment is needed, as seen from the experimental data of Kang, where the oscillation can begin even for ${\mathrm{Ra}}_{D}>400$. The heat transfer in this study’s computation overestimates Churchill and Chu’s correlation [3], but the graph is qualitatively similar to their result, showing a consistent error of 16.3%. In the high Ra region, the maximum overestimated error is 6% when ${\mathrm{Ra}}_{D}$ = $10^{4}$ in Morgan’s research [2]. However, the computational data consistently underpredict Kang’s experimental data by approximately 20%. Generally, the experiment seems far more unstable as there are many causes of disturbance, such as atmospheric perturbation in the laboratory, unremoved turbulence, the effect of surface roughness, radiative heat transfer, uncertainty of measurement, and failures in boundary temperature control.

Consequently, the lower limit of the present numerical method should be restricted to ${\mathrm{Ra}}_{D}>400$ due to the instability. However, as per Equation (16), if the gap ratio s/D is very small, the lower limit becomes far extended for ${\mathrm{Ra}}_{s}$.

Figure 4 shows the time history of the wall temperature compared to Kang’s experimental data in Figure 3, where the temperature is cooled to converge with the environmental air temperature. Although the initial wall temperature was the same, the experimental data indicate more rapid cooling as the negative slope of the wall temperature was steeper than in the case of the numerical approach. This means that additional heat transfer might occur at the test section, which may be attributed to the complex heat transfer from the atmosphere, or the failure of the adiabatic boundary condition in the measurement. The evaluation is very difficult in the case of natural convection because the flow speed is very low, but the effect of such an error cannot be neglected in such experiments.

### 3.2. Comparison with Experimental Data

In this research, the numerical data were also compared with the experimental data of Kang and Chang [10] to validate the proposed computational method. We used the range $2<{\mathrm{Ra}}_{s}<200$ based on the gap size of the fins. As shown in Figure 5, Nu, or the effect of convective heat transfer, increases as Ra becomes larger. In Figure 5, the trend of experimental data and numerical data is similar, but the numerical data underpredict Nu by approximately 16%, which is consistent with the previous comparison for a single tube.

A pivot point is observed at ${\mathrm{Ra}}_{s}\approx 100$, which is quite important for the following reason. Frequently, fins in heat exchangers are designed to increase the heat transfer. In general, larger fins should result in a higher Nu; however, the narrow gap for ${\mathrm{Ra}}_{s}<100$ can produce an adverse effect, given that D28, the largest fin diameter, shows the lowest heat transfer. The interval becomes large enough to remove the interference of the vertical fin walls when ${\mathrm{Ra}}_{s}>100$. Therefore, the Nusselt numbers are restored in the expected order in proportion to the size of the fins.

## 4. Results and Discussion

### 4.1. Velocity

Figure 6 shows the velocity distribution of air at the neutral plane between the fins for four fin diameters and two fin gaps. The air is accelerated vertically from downwards to upwards, but the patterns in (a) and (b) are different: in (a), the flow accelerates along the outer circumference of the fin, while in (b), it accelerates as the flow begins at the tube boundary. For the flow in (a), the air trapped in the gaps is not much affected by convection; rather, it circulates. Therefore, the flow can be regarded as such for an imaginary cylinder, wherein the outer fin diameter is equal to the cylinder’s diameter. As the fin diameter increases, the air velocity between the fins gradually decreases in Figure 6a while it increases in Figure 6b. For the air flow with low ${\mathrm{Ra}}_{s}$, the air pockets in the gap act as resistance, and the overall convective heat transfer decreases.

### 4.2. Temperature

Figure 7 shows the temperature distribution along the central cross section (i.e., the central axis of the tube). The temperature gradient of the air is lower than that at the base region of the fin (indicating a wider thermal boundary layer) in (a) with low Ra than those in (b), with high Ra. In Figure 7a, as the driven flow is suppressed at the gaps in Figure 6a, the cooling heat transfer is restricted to the region of the fin tips, whereas in Figure 7b, the convective heat transfer extends over the entire gap domain as the driven flow is activated from the side walls of the fins (see Figure 6b).

### 4.3. Fin Efficiency

When the heat transfer coefficient is uniform, the temperature of the fluid is constant, and there is one-dimensional heat conduction in the radial direction, the fin efficiency theory is as follows [12]:(17)$$\eta_{f}=\frac{2r}{m\left(r_{o}{}^{2}-r^{2}\right)}\left[\frac{K_{1}\left(mr\right)I_{1}\left(mr_{o}\right)-K_{1}\left(mr_{o}\right)I_{1}\left(mr\right)}{K_{1}\left(mr_{o}\right)I_{0}\left(mr\right)+K_{0}\left(mr\right)I_{1}\left(mr_{o}\right)}\right]$$

The fin efficiency is calculated from Schmidt’s approximation formula [12] and using the Bessel function as follows:(18)$$\eta_{f}=\frac{\tan h\left(mr\phi \right)}{mr\phi },\;$$
(19)$$m=\sqrt{2h/kt},\;\mathrm{and}$$
(20)$$\phi =\left(\frac{r_{o}}{r}-1\right)\left[1+0.35\ln\left(\frac{r_{o}}{r}\right)\right].\;$$
where m is the fin parameter, h is the heat transfer coefficient, k is the heat conductivity of the fin material, t is the thickness of the fin, $r_{o}$ is the radius of fin, and r is the radius of the tube.

Generally, the fin efficiency decreases as the fin diameter and heat transfer coefficient becomes large, but it increases as the fin thickness and heat conductivity increase. A thicker fin results in a small gap, and thus, the efficiency should increase.

Figure 8 shows the fin efficiency in the low Ra region (${\mathrm{Ra}}_{s}=15$). Fixing the fin thickness results in increased fin efficiency when the fin diameter becomes small and the fin gap is increased. As such, the fin efficiency increases when the shape of the heat exchanger becomes similar to that of a circular tube (or a larger gap exists), and thus, the effect of the fins degrades in the low Ra region.

## 5. Conclusionsigure

In this research, a series of numerical analyses were performed for a circular tube and 16 kinds of circular fin-tube heat exchangers. The numerical results were compared with experimental results and empirical correlations. The following are the salient points of this study:In a circular fin-tube heat exchanger, the heat transfer of natural convection can be simply expressed via a fin-tube model, and unsteady time marching was proposed to analyze time sweeping for various Ra values. This method is based on practical experiments with lumped temperature.Validation of the numerical analysis results via comparisons with empirical correlations such as those of Morgan [2] and Churchill and Chu [3] showed a limited range of errors. The proposed method underpredicted Kang’s experimental values by approximately 16%, but the overall trend coincided with his data. It was considered that the overestimation originated from the additional heat loss, except for the pure natural convection, which served as an artificial cause in the experiment. The accuracy of the natural convection data with the proposed numerical method can be guaranteed when ${\mathrm{Ra}}_{D}>400$, which is thought to be the limitation of numerical errors due to instability. Small differences in the temperature can affect the data severely. However, the proposed numerical simulation is far more stable than experiments where larger oscillations were observed for higher Ra values (${\mathrm{Ra}}_{D}\approx 3000)$The effective limit for a circular fin-tube heat exchanger is reached when ${\mathrm{Ra}}_{s}\geq 100$ because at low Ra values, the air gap between the fins is rarely affected by the natural convection from the outer air or stagnates when the fluid provides heat resistance. Therefore, the fin best serves its purpose when ${\mathrm{Ra}}_{s}$ exceeds 100. At low Ra values (${\mathrm{Ra}}_{s}=15$), shorter fins or a higher s/D ratio provide better efficiency.

## Figures and Tables

**Figure 1 entropy-22-00363-f001:**
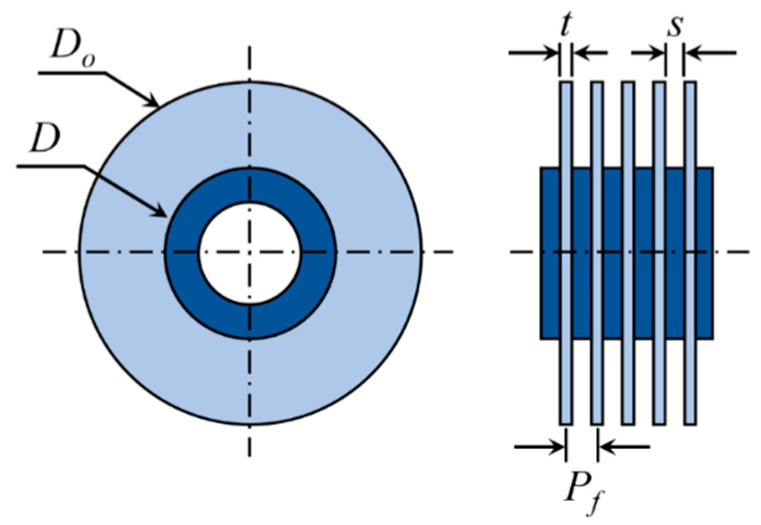
Schematic diagram of circular fin-tube heat exchanger studied in the present work.

**Figure 2 entropy-22-00363-f002:**
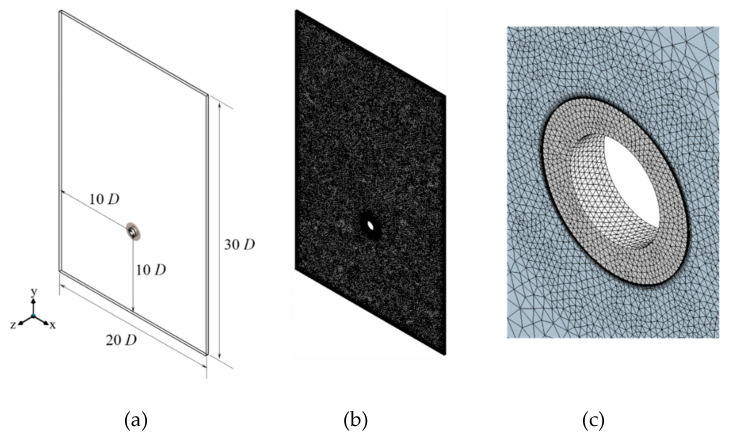
Numerical domain and grid in the present work: (**a**) computational domain, (**b**) grid (**c**) zoom in near the fin-tube.

**Figure 3 entropy-22-00363-f003:**
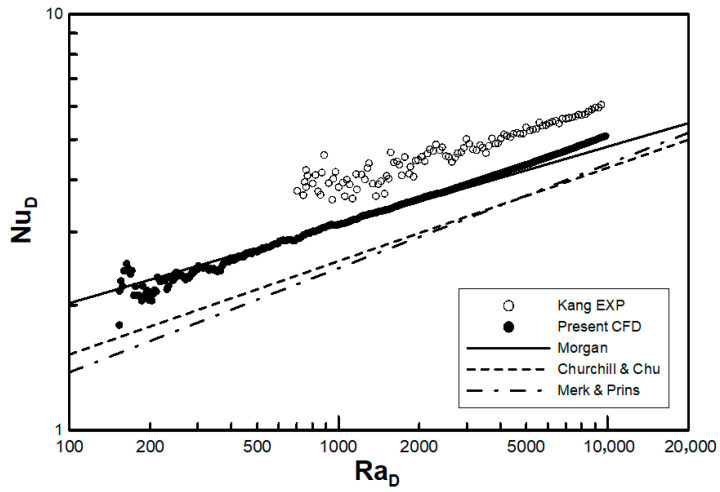
Comparison of Nusselt numbers with experimental correlation of circular tubes.

**Figure 4 entropy-22-00363-f004:**
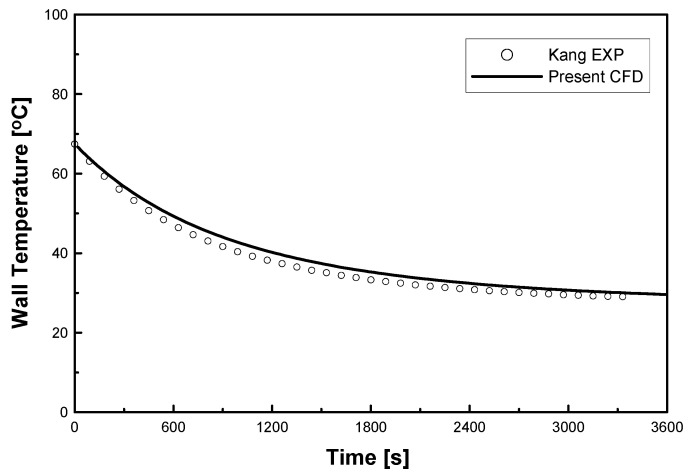
Comparison of wall temperature with experimental data of circular tubes.

**Figure 5 entropy-22-00363-f005:**
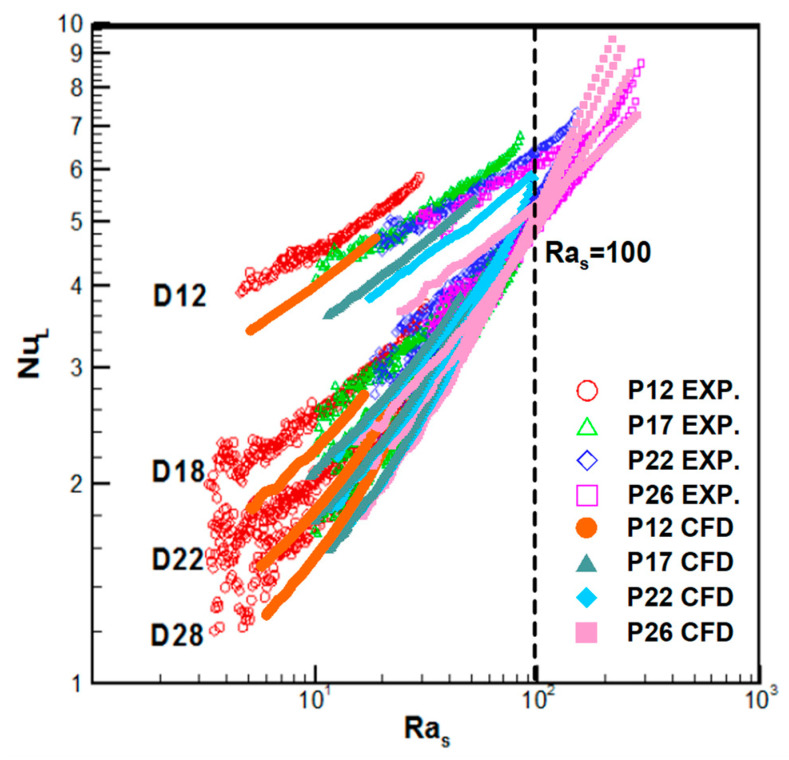
Comparison of Nu_L_ with experimental data of circular fin-tubes. (Experimental Data from Kang and Chang [10]).

**Figure 6 entropy-22-00363-f006:**
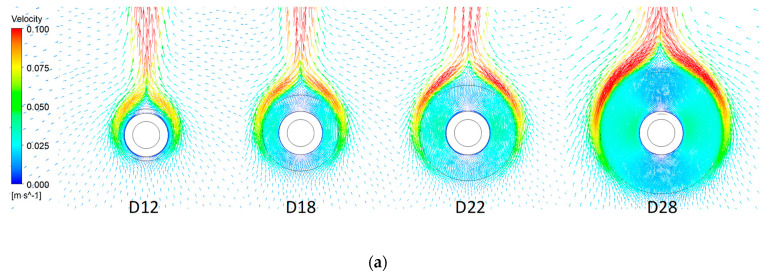
Velocity contours of the airflow in the present numerical experiment: (**a**) *s/D* = 0.119, ${\mathrm{Ra}}_{s}$ = 15; (**b**) *s/D* = 0.256, ${\mathrm{Ra}}_{s}$ = 150.

**Figure 7 entropy-22-00363-f007:**
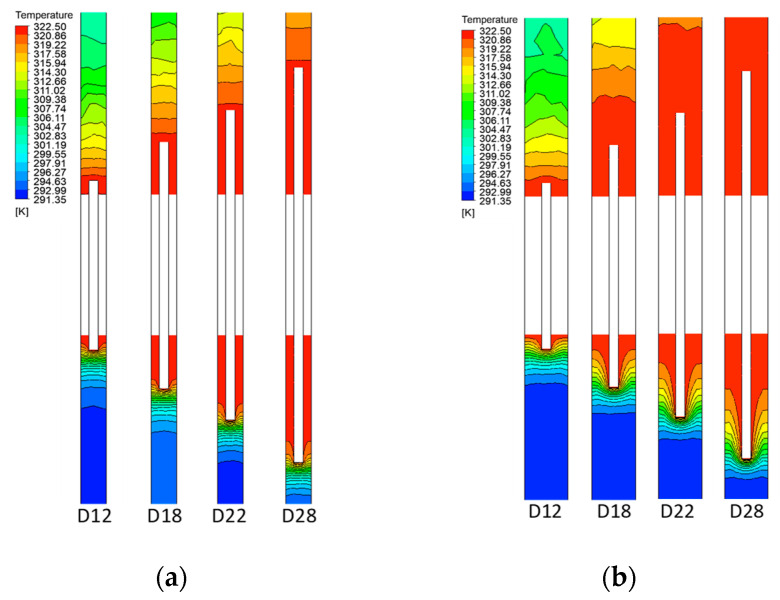
Isotherm lines for air-sides in the present numerical experiment: (**a**) *s/D* = 0.119, ${\mathrm{Ra}}_{s}$ = 15; (**b**) *s/D* = 0.256, ${\mathrm{Ra}}_{s}$ = 150.

**Figure 8 entropy-22-00363-f008:**
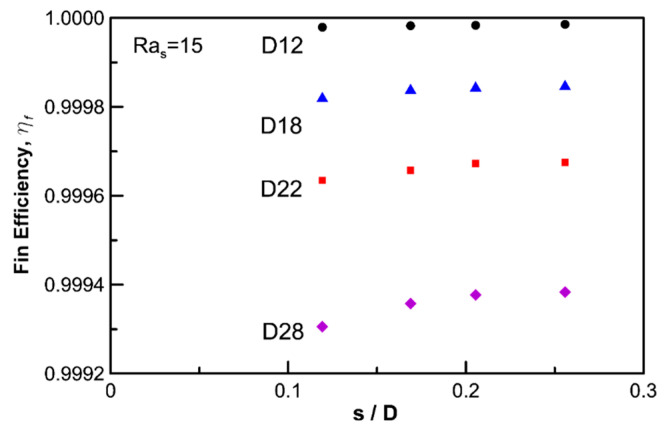
Variation of fin efficiency, *η_f_* with non-dimensional fin gap, *s*/*D*.

**Table 1 entropy-22-00363-t001:** Dimensions of the fin-tube heat exchangers tested in the present work.

Case		*D*	*D_o_*	*P_f_*	*t*	*D_o_*/*D*	*s*/*D*
**D12**	**P12**	15.88	19.1	2.89	1.0	1.20	0.119
	**P17**			3.68			0.169
	**P21**			4.26			0.205
	**P26**			5.06			0.256
**D18**	**P12**		27.8	2.89		1.75	0.119
	**P17**			3.68			0.169
	**P21**			4.26			0.205
	**P26**			5.06			0.256
**D22**	**P12**	15.88	34.9	2.89	1.0	2.20	0.119
	**P17**			3.68			0.169
	**P21**			4.26			0.205
	**P26**			5.06			0.256
**D28**	**P12**		44.5	2.89		2.80	0.119
	**P17**			3.68			0.169
	**P21**			4.26			0.205
	**P26**			5.06			0.256

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
