# Peer review of "Numerical Analysis on Natural Convection Heat Transfer in a Single Circular Fin-Tube Heat Exchanger (Part 1): Numerical Method"

_entropy, 2020, doi:10.3390/e22030363_

Round 1

Reviewer 1 Report

Comments are included in a pdf file.

Reviewer 2 Report

This study investigates numerically convective heat transfer around a circular fin disk connected to a heat exchanger tube. Several configurations are simulated in order to classify them with the Rayleigh numbers. 

Overvall, this manuscript is well written even though the introduction could be improved to highlight the novelty of this study. Some details regarding the numerical setup could also be added. Therefore, I would recommend publication if the authors take into account my comments.

Introduction: The literature does not provide enough details on the novelty of this study: This is true for experiments and CDF works reported. The reader is waiting for an in-depth analysis of pros and cons of all correlations presented.

  • Please define properly all dimensionless parameters.
  • p2, L68/ What do you mean by microscopic motion?
  • p2, L70/ The sentence of limitations of experiments should be more quantitative

Sec. 2: More details should be added to this section in order to fully understand the numerical methods used here.

  • p3, L81/ What is the thickness of the circular tube (of outer diameter D)?
  • In Tab.1, why don't you vary the thickness of the fin?
  • Eq. 4, You introduce a time step but I also see an initial and final conditions for temperature. Could you detail the calculation to reach this equation?
  • How do you obtain Eq. 7? Is there other way to define this reference length?
  • p4, L118/ Could you elaborate more on this numerical setup: time step, calculation time?  Reason for the mesh grid and mesh size?
  • Fig. 2a/ Please define here the (x,y,z) coordinates. Why do you use a periodic boundary conditions if one could inject some fluid in the inner tube so that a cooling process in the tube direction could happen

Sec. 3

  • p5, L152/ what kind of oscillations are you talking about? As I understood, at the beginning of the simulation, the delta of T is not high enough?
  • Figure 4/ You make some comparison with experiments. Where is located your probe and the thermocouple used in the experiment?
  • Fig. 5/ Superimpose your data with the one from experiments to really show the agreement.
  • Fig. 6/ According to you, how roughness could increase or decrease heat transfer in this configuration?
  • Fig. 7/ I wonder why the lowest Ra does not exhibit a natural convection induced flow due to the higher delta of T.
  • Figure 8/ You claim that the efficiency increases when the gap decreases. This is not in opposition with Fig. 6 and 7 (flow blockage)?

Round 2

Reviewer 1 Report

The authors have revised the manuscript in accordance with the comments in the first review. I can now recommend acceptance.